# Semi-Proximal Mirror-Prox
# for Nonsmooth Composite Minimization

**Niao He**
Georgia Institute of Technology
nhe6@gatech.edu

**Zaid Harchaoui**
NYU, Inria
firstname.lastname@nyu.edu

## Abstract

We propose a new first-order optimization algorithm to solve high-dimensional non-smooth composite minimization problems. Typical examples of such problems have an objective that decomposes into a non-smooth empirical risk part and a non-smooth regularization penalty. The proposed algorithm, called Semi-Proximal Mirror-Prox, leverages the saddle point representation of one part of the objective while handling the other part of the objective via linear minimization over the domain. The algorithm stands in contrast with more classical proximal gradient algorithms with smoothing, which require the computation of proximal operators at each iteration and can therefore be impractical for high-dimensional problems. We establish the theoretical convergence rate of Semi-Proximal Mirror-Prox, which exhibits the optimal complexity bounds, i.e. $O(1/\epsilon^2)$, for the number of calls to linear minimization oracle. We present promising experimental results showing the interest of the approach in comparison to competing methods.

## 1 Introduction

A wide range of machine learning and signal processing problems can be formulated as the minimization of a composite objective:

$$\min_{x \in X} F(x) := f(x) + \|\mathcal{B}x\| \tag{1}$$

where $X$ is closed and convex, $f$ is convex and can be either smooth, or nonsmooth yet enjoys a particular structure. The term $\|\mathcal{B}x\|$ defines a regularization penalty through a norm $\|\cdot\|$, and $x \mapsto \mathcal{B}x$ a linear mapping on a closed convex set $X$.

In many situations, the objective function $F$ of interest enjoys a favorable structure, namely a so-called *saddle point representation* [6, 11, 13]:

$$f(x) = \max_{z \in Z} \left\{ \langle x, Az \rangle - \psi(z) \right\} \tag{2}$$

where $Z$ is convex compact subset of a Euclidean space, and $\psi(\cdot)$ is a convex function. Sec. 4 will give several examples of such situations. Saddle point representations can then be leveraged to use first-order optimization algorithms.

The simple first option to minimize $F$ is using the so-called Nesterov smoothing technique [19] along with a proximal gradient algorithm [23], assuming that the proximal operator associated with $X$ is computationally tractable and cheap to compute. However, this is certainly not the case when considering problems with norms acting in the spectral domain of high-dimensional matrices, such as the matrix nuclear-norm [12] and structured extensions thereof [5, 2]. In the latter situation, another option is to use a smoothing technique now with a conditional gradient or Frank-Wolfe algorithm to minimize $F$, assuming that a *a linear minimization oracle* associated with $X$ is cheaper to compute than the proximal operator [6, 14, 24]. Neither option takes advantage of the composite structure of the objective (1) or handles the case when the linear mapping $\mathcal{B}$ is nontrivial.

**Contributions** Our goal is to propose a new first-order optimization algorithm, called Semi-Proximal Mirror-Prox, designed to solve the difficult non-smooth composite optimization problem (1), which does not require the exact computation of proximal operators. Instead, the Semi-Proximal Mirror-Prox relies upon i) Saddle point representability of $f$ (a less restricted role than Fenchel-type representation); ii) Linear minimization oracle associated with $\| \cdot \|$ in the domain $X$. While the saddle point representability of $f$ allows to cure the non-smoothness of $f$, the linear minimization over the domain $X$ allows to tackle the non-smooth regularization penalty $\| \cdot \|$. We establish the theoretical convergence rate of Semi-Proximal Mirror-Prox, which exhibits the *optimal complexity bounds*, i.e. $O(1/\epsilon^2)$, for the number of calls to linear minimization oracle. Furthermore, Semi-Proximal Mirror-Prox generalizes previously proposed approaches and improves upon them in special cases:

1. Case $\mathcal{B} \equiv 0$: Semi-Proximal Mirror-Prox does not require assumptions on favorable geometry of dual domain $Z$ or simplicity of $\psi(\cdot)$ in (2).

2. Case $\mathcal{B} = \mathbb{I}$: Semi-Proximal Mirror-Prox is competitive with previously proposed approaches [15, 24] based on smoothing techniques.

3. Case of non-trivial $\mathcal{B}$: Semi-Proximal Mirror-Prox is the first proximal-free or conditional-gradient-type optimization algorithm for (1).

**Related work** The Semi-Proximal Mirror-Prox algorithm belongs to the family of conditional gradient algorithms, whose most basic instance is the Frank-Wolfe algorithm for constrained smooth optimization using a linear minimization oracle; see [12, 1, 4]. Recently, in [6, 13], the authors consider constrained non-smooth optimization when the domain $Z$ has a "favorable geometry", *i.e.* the domain is amenable to proximal setups (favorable geometry), and establish a complexity bound with $O(1/\epsilon^2)$ calls to the linear minimization oracle. Recently, in [15], a method called conditional gradient sliding is proposed to solve similar problems, using a smoothing technique, with a complexity bound in $O(1/\epsilon^2)$ for the calls to the linear minimization oracle (LMO) and additionally a $O(1/\epsilon)$ bound for the linear operator evaluations. Actually, this $O(1/\epsilon^2)$ bound for the LMO complexity can be shown to be indeed *optimal* for conditional-gradient-type or LMO-based algorithms, when solving general[1] non-smooth convex problems [14].

However, these previous approaches are appropriate for objective with a non-composite structure. When applied to our problem (1), the smoothing would be applied to the objective taken as a whole, ignoring its composite structure. Conditional-gradient-type algorithms were recently proposed for composite objectives [7, 9, 26, 24, 16], but cannot be applied for our problem. In [9], $f$ is smooth and $\mathcal{B}$ is identity matrix, whereas in [24], $f$ is non-smooth and $\mathcal{B}$ is also the identity matrix. The proposed Semi-Proximal Mirror-Prox can be seen as a blend of the successful components resp. of the Composite Conditional Gradient algorithm [9] and the Composite Mirror-Prox [11], that enjoys the optimal complexity bound $O(1/\epsilon^2)$ on the total number of LMO calls, yet solves a broader class of convex problems than previously considered.

## 2 Framework and assumptions

We present here our theoretical framework, which hinges upon a smooth convex-concave saddle point reformulation of the norm-regularized non-smooth minimization (3). We shall use the following notations throughout the paper. For a given norm $\| \cdot \|$, we define the dual norm as $\|s\|_* = \max_{\|x\| \leq 1} \langle s, x \rangle$. For any $x \in \mathbf{R}^{m \times n}$, $\|x\|_2 = \|x\|_F = (\sum_{i=1}^m \sum_{j=1}^n |x_{ij}|^2)^{1/2}$.

**Problem** We consider the composite minimization problem

$$\text{Opt} = \min_{x \in X} \ f(x) + \|\mathcal{B}x\| \tag{3}$$

where $X$ is a closed convex set in the Euclidean space $E_x$; $x \mapsto \mathcal{B}x$ is a linear mapping from $X$ to $Y (\supset \mathcal{B}X)$, where $Y$ is a closed convex set in the Euclidean space $E_y$. We make two important assumptions on the function $f$ and the norm $\|\cdot\|$ defining the regularization penalty, explained below.

**Saddle Point Representation**  The non-smoothness of $f$ can be challenging to tackle. However, in many cases of interest, the function $f$ enjoys a favorable structure that allows to tackle it with smoothing techniques. We assume that $f(x)$ is a non-smooth convex function given by

$$f(x) = \max_{z \in Z} \Phi(x, z) \tag{4}$$

where $\Phi(x, z)$ is a smooth convex-concave function and $Z$ is a convex and compact set in the Euclidean space $E_z$. Such representation was introduced and developed in [6, 11, 13], for the purpose of non-smooth optimization. Saddle point representability can be interpreted as a general form of the smoothing-favorable structure of non-smooth functions used in the Nesterov smoothing technique [19]. Representations of this type are readily available for a wide family of "well-structured" nonsmooth functions $f$ (see Sec. 4 for examples ), and actually for all empirical risk functions with convex loss in machine learning, up to our knowledge.

**Composite Linear Minimization Oracle**  Proximal-gradient-type algorithms require the computation of a proximal operator at each iteration, i.e. $\min_{y \in Y} \left\{ \frac{1}{2}\|y\|_2^2 + \langle \eta, y \rangle + \alpha\|y\| \right\}$. For several cases of interest, described below, the computation of the proximal operator can be expensive or intractable. A classical example is the nuclear norm, whose proximal operator boils down to singular value thresholding, therefore requiring a full singular value decomposition. In contrast to the proximal operator, the linear minimization oracle can be much cheaper. The linear minimization oracle (LMO) is a routine which, given an input $\alpha > 0$ and $\eta \in E_y$, returns a point

$$\mathrm{LMO}(\eta, \alpha) := \operatorname*{argmin}_{y \in Y} \left\{ \langle \eta, y \rangle + \alpha\|y\| \right\} \tag{5}$$

In the case of nuclear-norm, the LMO only requires the computation of the leading pair of singular vectors, which is an order of magnitude faster in time-complexity.

**Saddle Point Reformulation.**  The crux of our approach is a smooth convex-concave saddle point reformulation of (3). After massaging the saddle-point reformulation, we consider the associated variational inequality, which provides the sufficient and necessary condition for an optimal solution to the saddle point problem [3, 4]. For any optimization problem with convex structure (including convex minimization, convex-concave saddle point problem, convex Nash equilibrium), the corresponding variational inequality is directly related to the accuracy certificate used to guarantee the accuracy of a solution to the optimization problem; see Sec. 2.1 in [11] and [18]. We shall present then an algorithm to solve the variational inequality established below, that exploits its particular structure.

Assuming that $f$ admits a saddle point representation (4), we write (3) in epigraph form

$$\mathrm{Opt} = \min_{x \in X, y \in Y, \tau \geq \|y\|} \max_{z \in Z} \left\{ \Phi(x, z) + \tau : y = \mathcal{B}x \right\}.$$

where $Y (\supset \mathcal{B}X)$ is a convex set. We can approximate Opt by

$$\widehat{\mathrm{Opt}} = \min_{x \in X, y \in Y, \tau \geq \|y\|} \max_{z \in Z, \|w\|_2 \leq 1} \left\{ \Phi(x, z) + \tau + \rho\langle y - \mathcal{B}x, w \rangle \right\}. \tag{6}$$

For properly selected $\rho > 0$, one has $\widehat{\mathrm{Opt}} = \mathrm{Opt}$ (see details in [11]). By introducing the variables $u := [x, y; z, w]$ and $v := \tau$, the variational inequality associated with the above saddle point problem is fully described by the domain

$$X_+ \quad = \quad \left\{ x_+ = [u; v] : x \in X, y \in Y, z \in Z, \|w\|_2 \leq 1, \tau \geq \|y\| \right\}$$

and the monotone vector field

$$F(x_+ = [u; v]) = [F_u(u); F_v],$$

where

$$F_u \left( u = \begin{bmatrix} x \\ y \\ z \\ w \end{bmatrix} \right) = \begin{bmatrix} \nabla_x \Phi(x, z) - \rho\mathcal{B}^T w \\ \rho w \\ -\nabla_z \Phi(x, z) \\ \rho(\mathcal{B}x - y) \end{bmatrix}, \qquad F_v(v = \tau) = 1.$$

In the next section, we present an efficient algorithm to solve this type of variational inequality, which enjoys a particular structure; we call such an inequality *semi-structured*.

# 3 Semi-Proximal Mirror-Prox for Semi-structured Variational Inequalities

Semi-structured variational inequalities (Semi-VI) enjoy a particular mixed structure, that allows to get the best of two worlds, namely the proximal setup (where the proximal operator can be computed) and the LMO setup (where the linear minimization oracle can be computed). Basically, the domain $X$ is decomposed as a Cartesian product over two sets $X = X_1 \times X_2$, such that $X_1$ admits a proximal-mapping while $X_2$ admits a linear minimization oracle. We now describe the main theoretical and algorithmic components of the Semi-Proximal Mirror-Prox algorithm, resp. in Sec. 3.1 and in Sec. 3.2, and finally describe the overall algorithm in Sec. 3.3.

## 3.1 Composite Mirror-Prox with Inexact Prox-mappings

We first present a new algorithm, which can be seen as an extension of the Composite Mirror Prox algorithm, denoted CMP for brevity, that allows inexact computation of prox-mappings and can solve a broad class of variational inequalites. The original Mirror Prox algorithm was introduced in [17] and was extended to composite settings in [11] assuming exact computations of prox-mappings.

**Structured Variational Inequalities.** We consider the variational inequality VI$(X, F)$:

$$\text{Find } x_* \in X : \langle F(x), x - x_* \rangle \geq 0, \forall x \in X$$

with domain $X$ and operator $F$ that satisfy the assumptions (**A**.1)–(**A**.4) below.

(**A**.1) Set $X \subset E_u \times E_v$ is closed convex and its projection $PX = \{u : x = [u; v] \in X\} \subset U$, where $U$ is convex and closed, $E_u, E_v$ are Euclidean spaces;

(**A**.2) The function $\omega(\cdot) : U \to \mathbf{R}$ is continuously differentiable and also 1-strongly convex w.r.t. some norm[2] $\| \cdot \|$. This defines the Bregman distance $V_u(u') = \omega(u') - \omega(u) - \langle \omega'(u), u' - u \rangle \geq \frac{1}{2}\|u' - u\|^2$.

(**A**.3) The operator $F(x = [u, v]) : X \to E_u \times E_v$ is monotone and of form $F(u, v) = [F_u(u); F_v]$ with $F_v \in E_v$ being a constant and $F_u(u) \in E_u$ satisfying the condition

$$\forall u, u' \in U : \|F_u(u) - F_u(u')\|_* \leq L\|u - u'\| + M$$

for some $L < \infty, M < \infty$;

(**A**.4) The linear form $\langle F_v, v \rangle$ of $[u; v] \in E_u \times E_v$ is bounded from below on $X$ and is coercive on $X$ w.r.t. $v$: whenever $[u^t; v^t] \in X$, $t = 1, 2, ...$ is a sequence such that $\{u^t\}_{t=1}^\infty$ is bounded and $\|v^t\|_2 \to \infty$ as $t \to \infty$, we have $\langle F_v, v^t \rangle \to \infty, t \to \infty$.

The quality of an iterate, in the course of the algorithm, is measured through the so-called dual gap function

$$\epsilon_{\text{VI}}(x|X, F) = \sup_{y \in X} \langle F(y), x - y \rangle .$$

We give in Appendix A a refresher on dual gap functions, for the reader's convenience. We shall establish the complexity bounds in terms of this dual gap function for our algorithm, which directly provides an accuracy certificate along the iterations. However, we first need to define what we mean by an inexact prox-mapping.

$\epsilon$**-Prox-mapping** Inexact proximal mappings were recently considered in the context of accelerated proximal gradient algorithms [25]. The definition we give below is more general, allowing for non-Euclidean proximal-mappings.

We introduce here the notion of $\epsilon$-prox-mapping for $\epsilon \geq 0$. For $\xi = [\eta; \zeta] \in E_u \times E_v$ and $x = [u; v] \in X$, let us define the subset $P_x^\epsilon(\xi)$ of $X$ as

$$P_x^\epsilon(\xi) = \{\widehat{x} = [\widehat{u}; \widehat{v}] \in X : \langle \eta + \omega'(\widehat{u}) - \omega'(u), \widehat{u} - s \rangle + \langle \zeta, \widehat{v} - w \rangle \leq \epsilon \; \forall [s; w] \in X\}.$$

When $\epsilon = 0$, this reduces to the exact prox-mapping, in the usual setting, that is

$$P_x(\xi) = \underset{[s;w] \in X}{\text{Argmin}} \{\langle \eta, s \rangle + \langle \zeta, w \rangle + V_u(s)\} .$$

When $\epsilon > 0$, this yields our definition of an inexact prox-mapping, with inexactness parameter $\epsilon$. Note that for any $\epsilon \geq 0$, the set $P_x^\epsilon(\xi = [\eta; \gamma F_v])$ is well defined whenever $\gamma > 0$. The Composite Mirror Prox with inexact prox-mappings is outlined in Algorithm 1.

---

**Algorithm 1** Composite Mirror Prox Algorithm (CMP) for $\mathrm{VI}(X, F)$

---

**Input:** stepsizes $\gamma_t > 0$, inexactness $\epsilon_t \geq 0$, $t = 1, 2, \ldots$
Initialize $x^1 = [u^1; v^1] \in X$
**for** $t = 1, 2, \ldots, T$ **do**
$$\begin{aligned} y^t := [\widehat{u}^t; \widehat{v}^t] &\in P_{x^t}^{\epsilon_t}(\gamma_t F(x^t)) = P_{x^t}^{\epsilon_t}(\gamma_t[F_u(u^t); F_v]) \\ x^{t+1} := [u^{t+1}; v^{t+1}] &\in P_{x^t}^{\epsilon_t}(\gamma_t F(y^t)) = P_{x^t}^{\epsilon_t}(\gamma_t[F_u(\widehat{u}^t); F_v]) \end{aligned} \tag{7}$$
**end for**
**Output:** $\overline{x}_T := [\overline{u}_T; \overline{v}_T] = \left(\sum_{t=1}^T \gamma_t\right)^{-1} \sum_{t=1}^T \gamma_t y^t$

---

The proposed algorithm is a non-trivial extension of the Composite Mirror Prox with *exact prox-mappings*, both from a theoretical and algorithmic point of views. We establish below the theoretical convergence rate; see Appendix B for the proof.

**Theorem 3.1.** *Assume that the sequence of step-sizes $(\gamma_t)$ in the CMP algorithm satisfy*

$$\sigma_t := \gamma_t \langle F_u(\widehat{u}^t) - F_u(u^t), \widehat{u}^t - u^{t+1} \rangle - V_{\widehat{u}^t}(u^{t+1}) - V_{u^t}(\widehat{u}^t) \leq \gamma_t^2 M^2, \quad t = 1, 2, \ldots, T \ . \tag{8}$$

*Then, denoting $\Theta[X] = \sup_{[u;v] \in X} V_{u^1}(u)$, for a sequence of inexact prox-mappings with inexactness $\epsilon_t \geq 0$, we have*

$$\epsilon_{\mathrm{VI}}(\overline{x}_T | X, F) := \sup_{x \in X} \ \langle F(x), \overline{x}_T - x \rangle \leq \frac{\Theta[X] + M^2 \sum_{t=1}^T \gamma_t^2 + 2 \sum_{t=1}^T \epsilon_t}{\sum_{t=1}^T \gamma_t}. \tag{9}$$

**Remarks.** Note that the assumption on the sequence of step-sizes $(\gamma_t)$ is clearly satisfied when $\gamma_t \leq (\sqrt{2}L)^{-1}$. When $M = 0$ (which is essentially the case for the problem described in Section 2), it suffices as long as $\gamma_t \leq L^{-1}$. When $(\epsilon_t)$ is summable, we achieve the same $O(1/T)$ convergence rate as when there is no error. If $(\epsilon_t)$ decays with a rate of $O(1/t)$, then the overall convergence is only affected by a $\log(T)$ factor. Convergence results on the sequence of projections of $(\overline{x}_T)$ onto $X_1$ when $F$ stems from saddle point problem $\min_{x^1 \in X_1} \sup_{x^2 \in X_2} \Phi(x^1, x^2)$ is established in Appendix B.

The theoretical convergence rate established in Theorem 3.1 and Corollary B.1 generalizes the previous result established in Corollary 3.1 in [11] for CMP with exact prox-mappings. Indeed, when exact prox-mappings are used, we recover the result of [11]. When inexact prox-mappings are used, the errors due to the inexactness of the prox-mappings accumulate and is reflected in (9) and (37).

### 3.2 Composite Conditional Gradient

We now turn to a variant of the composite conditional gradient algorithm, denoted CCG, tailored for a particular class of problems, which we call *smooth semi-linear problems*. The composite conditional gradient algorithm was first introduced in [9] and also developed in [21]. We present an extension here which turns to be well-suited for sub-problems that will be solved in Sec. 3.3.

**Minimizing Smooth Semi-linear Functions.** We consider the smooth semi-linear problem

$$\min_{x=[u;v] \in X} \left\{ \phi^+(u, v) = \phi(u) + \langle \theta, v \rangle \right\} \tag{10}$$

represented by the pair $(X; \phi^+)$ such that the following assumptions are satisfied. We assume that

i) $X \subset E_u \times E_v$ is closed convex and its projection $PX$ on $E_u$ belongs to $U$, where $U$ is convex and compact;

ii) $\phi(u) : U \to \mathbf{R}$ is a convex continuously differentiable function, and there exist $1 < \kappa \leq 2$ and $L_0 < \infty$ such that

$$\phi(u') \leq \phi(u) + \langle \nabla\phi(u), u' - u \rangle + \frac{L_0}{\kappa} \|u' - u\|^\kappa \ \forall u, u' \in U; \tag{11}$$

iii) $\theta \in E_v$ is such that every linear function on $E_u \times E_v$ of the form

$$[u; v] \mapsto \langle \eta, u \rangle + \langle \theta, v \rangle \tag{12}$$

with $\eta \in E_u$ attains its minimum on $X$ at some point $x[\eta] = [u[\eta]; v[\eta]]$; we have at our disposal a *Composite Linear Minimization Oracle* (LMO) which, given on input $\eta \in E_u$, returns $x[\eta]$.

---

**Algorithm 2** Composite Conditional Gradient Algorithm $\mathbf{CCG}(X, \phi(\cdot), \theta; \epsilon)$

---

**Input:** accuracy $\epsilon > 0$ and $\gamma_t = 2/(t+1), t = 1, 2, \dots$
Initialize $x^1 = [u^1; v^1] \in X$
**for** $t = 1, 2, \dots$ **do**
    Compute $\delta_t = \langle g_t, u^t - u^t[g_t] \rangle + \langle \theta, v^t - v^t[g_t] \rangle$, where $g_t = \nabla\phi(u^t)$;
    **if** $\delta_t \leq \epsilon$ **then**
        Return $x^t = [u^t; v^t]$
    **else**
        Find $x^{t+1} = [u^{t+1}; v^{t+1}] \in X$ such that $\phi^+(x^{t+1}) \leq \phi^+ (x^t + \gamma_t(x^t[g_t] - x^t))$
    **end if**
**end for**

---

The algorithm is outlined in Algorithm 2. Note that CCG works essentially *as if* there were no $v$-component at all. The CCG algorithm enjoys a convergence rate in $O(t^{-(\kappa-1)})$ in the evaluations of the function $\phi^+$, and the accuracy certificates $(\delta_t)$ enjoy the same rate $O(t^{-(\kappa-1)})$ as well.

**Proposition 3.1.** *Denote D the $\|\cdot\|$-diameter of U. When solving problems of type (10), the sequence of iterates $(x^t)$ of CCG satisfies*

$$\phi^+(x^t) - \min_{x \in X} \phi^+(x) \leq \frac{2L_0 D^\kappa}{\kappa(3-\kappa)} \left( \frac{2}{t+1} \right)^{\kappa-1}, \ t \geq 2 \tag{13}$$

*In addition, the accuracy certificates $(\delta_t)$ satisfy*

$$\min_{1 \leq s \leq t} \delta_s \leq O(1) L_0 D^\kappa \left( \frac{2}{t+1} \right)^{\kappa-1}, \ t \geq 2 \tag{14}$$

### 3.3 Semi-Proximal Mirror-Prox for Semi-structured Variational Inequality

We now give the full description of a special class of variational inequalities, called *semi-structured variational inequalities*. This family of problems encompasses both cases that we discussed so far in Section 3.1 and 3.2. But most importantly, it also covers many other problems that do not fall into these two regimes and in particular, our essential problem of interest (3).

**Semi-structured Variational Inequalities.** The class of semi-structured variational inequalities allows to go beyond Assumptions $(\mathbf{A}.1) - (\mathbf{A}.4)$, by assuming more structure. This structure is consistent with what we call a *semi-proximal* setup, which encompasses both the regular *proximal setup* and the regular *linear minimization setup* as special cases. Indeed, we consider variational inequality $\mathrm{VI}(X, F)$ that satisfies, in addition to Assumptions $(\mathbf{A}.1) - (\mathbf{A}.4)$, the following assumptions:

(**S**.1) *Proximal setup for $X$*: we assume that $E_u = E_{u_1} \times E_{u_2}$, $E_v = E_{v_1} \times E_{v_2}$, and $U \subset U_1 \times U_2$, $X = X_1 \times X_2$ with $X_i \in E_{u_i} \times E_{v_i}$ and $P_i X = \{u_i : [u_i; v_i] \in X_i\} \subset U_i$ for $i = 1, 2$, where $U_1$ is convex and closed, $U_2$ is convex and compact. We also assume that $\omega(u) = \omega_1(u_1) + \omega_2(u_2)$ and $\|u\| = \|u_1\|_{E_{u_1}} + \|u_2\|_{E_{u_2}}$, with $\omega_2(\cdot) : U_2 \to \mathbf{R}$ continuously differentiable such that

$$\omega_2(u_2') \leq \omega_2(u_2) + \langle \nabla\omega_2(u_2), u_2' - u_2 \rangle + \frac{L_0}{\kappa} \|u_2' - u_2\|_{E_{u_2}}^\kappa, \forall u_2, u_2' \in U_2;$$

for a particular $1 < \kappa \leq 2$ and $L_0 < \infty$. Furthermore, we assume that the $\| \cdot \|_{E_{u_2}}$-diameter of $U_2$ is bounded by some $D > 0$.

(**S**.2) *Partition of $F$*: the operator $F$ induced by the above partition of $X_1$ and $X_2$ can be written as

$$F(x) = [F_u(u); F_v] \text{ with } F_u(u) = [F_{u_1}(u_1, u_2); F_{u_2}(u_1, u_2)], F_v = [F_{v_1}; F_{v_2}].$$

(**S**.3) *Proximal mapping on $X_1$*: we assume that for any $\eta_1 \in E_{u_1}$ and $\alpha > 0$, we have at our disposal easy-to-compute prox-mappings of the form,

$$\text{Prox}_{\omega_1}(\eta_1, \alpha) := \underset{x_1 = [u_1; v_1] \in X_1}{\text{argmin}} \{\omega_1(u_1) + \langle \eta_1, u_1 \rangle + \alpha \langle F_{v_1}, v_1 \rangle\}.$$

(**S**.4) *Linear minimization oracle for $X_2$*: we assume that we we have at our disposal Composite Linear Minimization Oracle (LMO), which given any input $\eta_2 \in E_{u_2}$ and $\alpha > 0$, returns an optimal solution to the minimization problem with linear form, that is,

$$\text{LMO}(\eta_2, \alpha) := \underset{x_2 = [u_2; v_2] \in X_2}{\text{argmin}} \{\langle \eta_2, u_2 \rangle + \alpha \langle F_{v_2}, v_2 \rangle\}.$$

**Semi-proximal setup**  We denote such problems as Semi-VI$(X, F)$. On the one hand, when $U_2$ is a singleton, we get the *full-proximal setup*. On the other hand, when $U_1$ is a singleton, we get the *full linear-minimization-oracle setup* (full LMO setup). The *semi-proximal setup* allows to cover both setups and all the ones in between as well.

**The Semi-Proximal Mirror-Prox algorithm.**  We finally present here our main contribution, the Semi-Proximal Mirror-Prox algorithm, which solves the semi-structured variational inequality under $(\mathbf{A}.1) - (\mathbf{A}.4)$ and $(\mathbf{S}.1) - (\mathbf{S}.4)$. The Semi-Proximal Mirror-Prox algorithm blends both CMP and CCG. Basically, for sub-domain $X_2$ given by LMO, instead of computing exactly the prox-mapping, we mimick inexactly the prox-mapping via a conditional gradient algorithm in the Composite Mirror Prox algorithm. For the sub-domain $X_1$, we compute the prox-mapping as it is.

---

**Algorithm 3 Semi-Proximal Mirror-Prox** Algorithm for Semi-VI$(X, F)$

---

**Input:** stepsizes $\gamma_t > 0$, accuracies $\epsilon_t \geq 0$, $t = 1, 2, \ldots$
[1] Initialize $x^1 = [x_1^1; x_2^1] \in X$, where $x_1^1 = [u_1^1; v_1^1]; x_2^1 = [u_2^1,; v_2^1]$.
**for** $t = 1, 2, \ldots, T$ **do**
  [2] Compute $y^t = [y_1^t; y_2^t]$ that

$$
\begin{aligned}
y_1^t := [\widehat{u}_1^t; \widehat{v}_1^t] &= \text{Prox}_{\omega_1}(\gamma_t F_{u_1}(u_1^t, u_2^t) - \omega_1'(u_1^t), \gamma_t) \\
y_2^t := [\widehat{u}_2^t; \widehat{v}_2^t] &= \mathbf{CCG}(X_2, \omega_2(\cdot) + \langle \gamma_t F_{u_2}(u_1^t, u_2^t) - \omega_2'(u_2^t), \cdot \rangle, \gamma_t F_{v_2}; \epsilon_t)
\end{aligned}
$$

  [3] Compute $x^{t+1} = [x_1^{t+1}; x_2^{t+1}]$ that

$$
\begin{aligned}
x_1^{t+1} := [u_1^{t+1}; v_1^{t+1}] &= \text{Prox}_{\omega_1}(\gamma_t F_{u_1}(\widehat{u}_1^t, \widehat{u}_2^t) - \omega_1'(u_1^t), \gamma_t) \\
x_2^{t+1} := [u_2^{t+1}; v_2^{t+1}] &= \mathbf{CCG}(X_2, \omega_2(\cdot) + \langle \gamma_t F_{u_2}(\widehat{u}_1^t, \widehat{u}_2^t) - \omega_2'(u_2^t), \cdot \rangle, \gamma_t F_{v_2}; \epsilon_t)
\end{aligned}
$$

**end for**
**Output:** $\bar{x}_T := [\bar{u}_T; \bar{v}_T] = \left(\sum_{t=1}^T \gamma_t\right)^{-1} \sum_{t=1}^T \gamma_t y^t$

---

At step $t$, we first update $y_1^t = [\widehat{u}_1^t; \widehat{v}_1^t]$ by computing the exact prox-mapping and build $y_2^t = [\widehat{u}_2^t; \widehat{v}_2^t]$ by running the composite conditional gradient algorithm to problem (10) specifically with

$$X = X_2, \phi(\cdot) = \omega_2(\cdot) + \langle \gamma_t F_{u_2}(u_1^t, u_2^t) - \omega_2'(u_2^t), \cdot \rangle, \text{ and } \theta = \gamma_t F_{v_2},$$

until $\delta(y_2^t) = \max_{y_2 \in X_2} \langle \nabla \phi^+(y_2^t), y_2^t - y_2 \rangle \leq \epsilon_t$. We then build $x_1^{t+1} = [u_1^{t+1}; v_1^{t+1}]$ and $x_2^{t+1} = [u_2^{t+1}; v_2^{t+1}]$ similarly except this time taking the value of the operator at point $y^t$. Combining the results in Theorem 3.1 and Proposition 3.1, we arrive at the following complexity bound.

**Proposition 3.2.** *Under the assumption $(\mathbf{A}.1) - (\mathbf{A}.4)$ and $(\mathbf{S}.1) - (\mathbf{S}.4)$ with $M = 0$, and choice of stepsize $\gamma_t = L^{-1}, t = 1, \ldots, T$, for the outlined algorithm to return an $\epsilon$-solution to the variational inequality $VI(X, F)$, the total number of Mirror Prox steps required does not exceed*

$$\text{Total number of steps} = O(1)\frac{L\Theta[X]}{\epsilon}$$

*and the total number of calls to the Linear Minimization Oracle does not exceed*

$$\mathcal{N} = O(1) \left(\frac{L_0 L^\kappa D^\kappa}{\epsilon^\kappa}\right)^{\frac{1}{\kappa - 1}} \Theta[X].$$

*In particular, if we use Euclidean proximal setup on $U_2$ with $\omega_2(\cdot) = \frac{1}{2}\|x_2\|^2$, which leads to $\kappa = 2$ and $L_0 = 1$, then the number of LMO calls does not exceed $\mathcal{N} = O(1)\left(L^2 D^2(\Theta[X_1] + D^2)\right)/\epsilon^2$.*

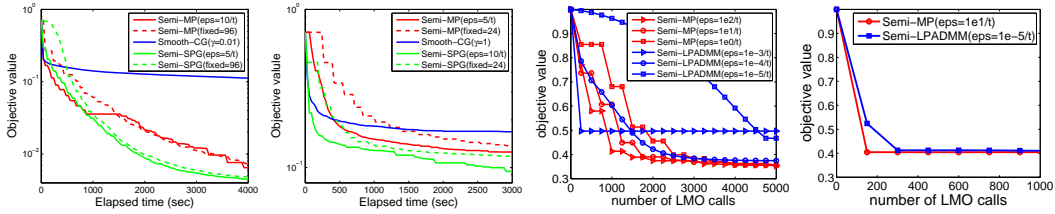

Figure 1: Robust collaborative filtering and link prediction: objective function vs elapsed time. From left to right: (a) MovieLens100K; (b) MovieLens1M; (c) Wikivote (1024); (d) Wikivote (full)

**Discussion**   The proposed Semi-Proximal Mirror-Prox algorithm enjoys the *optimal complexity bounds*, i.e. $O(1/\epsilon^2)$, in the number of calls to LMO; see [14] for the optimal complexity bounds for general non-smooth optimization with LMO. Consequently, when applying the algorithm to the variational reformulation of the problem of interest (3), we are able to get an $\epsilon$-optimal solution within at most $O(1/\epsilon^2)$ LMO calls.   Thus, Semi-Proximal Mirror-Prox generalizes previously proposed approaches and improves upon them in special cases of problem (3); see Appendix D.2.

## 4   Experiments

We report the experimental results obtained with the proposed Semi-Proximal Mirror-Prox, denoted **Semi-MP** here, and competing algorithms.  We consider two different applications: i) robust collaborative filtering for movie recommendation; ii) link prediction for social network analysis.  For i), we compare to two competing approaches: a) smoothing conditional gradient proposed in [24] (denoted Smooth-CG); b) smoothing proximal gradient [20, 5] equipped with semi-proximal setup (Semi-SPG).  For ii), we compare to Semi-LPADMM, using [22] equipped with semi-proximal setup. Additional experiments and implementation details are given in Appendix E.

**Robust collaborative filtering**   We consider the collaborative filtering problem, with a nuclear-norm regularization penalty and an $\ell_1$-loss function.  We run the above three algorithms on the the small and medium MovieLens datasets. The small-size dataset consists of 943 users and 1682 movies with about 100K ratings, while the medium-size dataset consists of 3952 users and 6040 movies with about 1M ratings. We follow [24] to set the regularization parameters. In Fig. 1, we can see that Semi-MP clearly outperforms Smooth-CG, while it is competitive with Semi-SPG.

**Link prediction**   We consider now the link prediction problem, where the objective consists a hinge-loss for the empirical risk part and multiple regularization penalties, namely the $\ell_1$-norm and the nuclear-norm.  For this example, applying the Smooth-CG or Semi-SPG would require two smooth approximations, one for hinge loss term and one for $\ell_1$ norm term. Therefore, we consider an alternative approach, Semi-LPADMM, where we apply the linearized preconditioned ADMM algorithm [22] by solving proximal mapping through conditional gradient routines. Up to our knowledge, ADMM with early stopping is not fully theoretically analyzed in literature. However, intuitively, as long as the error is controlled sufficiently, such variant of ADMM should converge.

We conduct experiments on a binary social graph data set called Wikivote, which consists of 7118 nodes and 103747 edges. Since the computation cost of these two algorithms mainly come from the LMO calls, we present in below the performance in terms of number of LMO calls. For the first set of experiments, we select top 1024 highest degree users from Wikivote and run the two algorithms on this small dataset with different strategies for the inner LMO calls.

In Fig. 1, we observe that the Semi-MP is less sensitive to the inner accuracies of prox-mappings compared to the ADMM variant, which sometimes stops progressing if the prox-mappings of early iterations are not solved with sufficient accuracy. The results on the full dataset corroborate the fact that Semi-MP outperforms the semi-proximal variant of the ADMM algorithm.

**Acknowledgments**

The authors would like to thank A. Juditsky and A. Nemirovski for fruitful discussions. This work was supported by NSF Grant CMMI-1232623, LabEx Persyval-Lab (ANR-11-LABX-0025), project "Titan" (CNRS-Mastodons), project "Macaron" (ANR-14-CE23-0003-01), the MSR-Inria joint centre, and the Moore-Sloan Data Science Environment at NYU.

## Footnotes

[1]Related research extended such approaches to stochastic or online settings [10, 8, 15]; such settings are beyond the scope of this work.

[2]There is a slight abuse of notation here. The norm here is not the same as the one in problem (3)

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
