[Supplementary Material]

In this Appendix, we provide additional material on variational inequalities and non-smooth optimization algorithms, give the proofs on the main theorems, and provide additional information regarding the competing algorithms based on smoothing techniques and the implementation details for different models.

## A Preliminaries: Variational Inequalities and Accuracy Certificates

For the reader's convenience, we recall here the relationship between variational inequalities, accuracy certificates, and execution protocols, for non-smooth optimization algorithms. The exposition below is directly taken from [11], and recalled here for the reader's convenience.

**Execution protocols and accuracy certificates.** Let $X$ be a nonempty closed convex set in a Euclidean space $E$ and $F(x) : X \to E$ be a vector field.

Suppose that we process $(X, F)$ by an algorithm which generates a sequence of search points $x_t \in X$, $t = 1, 2, ...$, and computes the vectors $F(x_t)$, so that after $t$ steps we have at our disposal $t$-*step execution protocol* $\mathcal{I}_t = \{x_\tau, F(x_\tau)\}_{\tau=1}^t$. By definition, an *accuracy certificate* for this protocol is simply a collection $\lambda^t = \{\lambda_\tau^t\}_{\tau=1}^t$ of nonnegative reals summing up to 1. We associate with the protocol $\mathcal{I}_t$ and accuracy certificate $\lambda^t$ two quantities as follows:

- *Approximate solution* $x^t(\mathcal{I}_t, \lambda^t) := \sum_{\tau=1}^t \lambda_\tau^t x_\tau$, which is a point of $X$;
- *Resolution* $\mathrm{Res}(X' | \mathcal{I}_t, \lambda^t)$ *on a subset* $X' \neq \emptyset$ of $X$ given by

$$\mathrm{Res}(X' | \mathcal{I}_t, \lambda^t) = \sup_{x \in X'} \sum_{\tau=1}^t \lambda_\tau^t \langle F(x_\tau), x_\tau - x \rangle. \tag{15}$$

The role of those notions for non-smooth optimization is explained below.

**Variational inequalities.** Assume that $F$ is *monotone*, i.e., VI(X,F)

$$\langle F(x) - F(y), x - y \rangle \geq 0, \quad \forall x, y \in X . \tag{16}$$

Our goal is to approximate a weak solution to the variational inequality (v.i.) $\mathrm{VI}(X, F)$ associated with $(X, F)$. A weak solution is defined as a point $x_* \in X$ such that

$$\langle F(y), y - x_* \rangle \geq 0 \ \forall y \in X. \tag{17}$$

A natural (in)accuracy measure of a candidate weak solution $x \in X$ to $\mathrm{VI}(X, F)$ is the *dual gap function*

$$\epsilon_{\mathrm{VI}}(x | X, F) = \sup_{y \in X} \langle F(y), x - y \rangle \tag{18}$$

This inaccuracy is a convex nonnegative function which vanishes exactly at the set of weak solutions to the $\mathrm{VI}(X, F)$.

**Proposition A.1.** *For every $t$, every execution protocol $\mathcal{I}_t = \{x_\tau \in X, F(x_\tau)\}_{\tau=1}^t$ and every accuracy certificate $\lambda^t$ one has $x^t := x^t(\mathcal{I}_t, \lambda^t) \in X$. Besides this, assuming $F$ monotone, for every closed convex set $X' \subset X$ such that $x^t \in X'$ one has*

$$\epsilon_{\mathrm{VI}}(x^t | X', F) \leq \mathrm{Res}(X' | \mathcal{I}_t, \lambda^t). \tag{19}$$

*Proof.* Indeed, $x^t$ is a convex combination of the points $x_\tau \in X$ with coefficients $\lambda_\tau^t$, whence $x^t \in X$. With $X'$ as in the premise of Proposition, we have

$$\forall y \in X' : \langle F(y), x^t - y \rangle = \sum_{\tau=1}^t \lambda_\tau^t \langle F(y), x_\tau - y \rangle \leq \sum_{\tau=1}^t \lambda_\tau^t \langle F(x_\tau), x_\tau - y \rangle \leq \mathrm{Res}(X' | \mathcal{I}_t, \lambda^t),$$

where the first $\leq$ is due to monotonicity of $F$. □

**Convex-concave saddle point problems.** Now let $X = X_1 \times X_2$, where $X_i$ is a closed convex subset in Euclidean space $E_i$, $i = 1, 2$, and $E = E_1 \times E_2$, and let $\Phi(x^1, x^2) : X_1 \times X_2 \to \mathbf{R}$ be a locally Lipschitz continuous function which is convex in $x^1 \in X_1$ and concave in $x^2 \in X_2$. $X_1, X_2, \Phi$ give rise to the saddle point problem

$$\text{SadVal} = \min_{x^1 \in X_1} \max_{x^2 \in X_2} \Phi(x^1, x^2), \tag{20}$$

two induced convex optimization problems

$$
\begin{aligned}
\text{Opt}(P) &= \min_{x^1 \in X_1} \left[ \overline{\Phi}(x^1) = \sup_{x^2 \in X_2} \Phi(x^1, x^2) \right] \quad (P) \\
\text{Opt}(D) &= \max_{x^2 \in X_2} \left[ \underline{\Phi}(x^2) = \inf_{x^1 \in X_1} \Phi(x^1, x^2) \right] \quad (D)
\end{aligned}
\tag{21}
$$

and a vector field $F(x^1, x^2) = [F_1(x^1, x^2); F_2(x^1, x^2)]$ specified (in general, non-uniquely) by the relations

$$\forall(x^1, x^2) \in X_1 \times X_2 : F_1(x^1, x^2) \in \partial_{x^1} \Phi(x^1, x^2), \ F_2(x^1, x^2) \in \partial_{x^2}[-\Phi(x^1, x^2)].$$

It is well known that $F$ is monotone on $X$, and that weak solutions to the $\text{VI}(X, F)$ are exactly the saddle points of $\Phi$ on $X_1 \times X_2$. These saddle points exist if and only if $(P)$ and $(D)$ are solvable with equal optimal values, in which case the saddle points are exactly the pairs $(x^1_*, x^2_*)$ comprised by optimal solutions to $(P)$ and $(D)$. In general, $\text{Opt}(P) \geq \text{Opt}(D)$, with equality definitely taking place when at least one of the sets $X_1, X_2$ is bounded; if both are bounded, saddle points do exist. To avoid unnecessary complications, from now on, when speaking about a convex-concave saddle point problem, we assume that the problem is *proper*, meaning that $\text{Opt}(P)$ and $\text{Opt}(D)$ are reals; this definitely is the case when $X$ is bounded.

A natural (in)accuracy measure for a candidate $x = [x^1; x^2] \in X_1 \times X_2$ to the role of a saddle point of $\Phi$ is the quantity

$$
\begin{aligned}
\epsilon_{\text{Sad}}(x | X_1, X_2, \Phi) &= \overline{\Phi}(x^1) - \underline{\Phi}(x^2) \\
&= [\overline{\Phi}(x^1) - \text{Opt}(P)] + [\text{Opt}(D) - \underline{\Phi}(x^2)] + [\text{Opt}(P) - \text{Opt}(D)]
\end{aligned}
\tag{22}
$$

This inaccuracy is nonnegative. It is the sum of the duality gap $\text{Opt}(P) - \text{Opt}(D)$ (always nonnegative and vanishing when one of the sets $X_1, X_2$ is bounded), and the two inaccuracies resp. of $x^1$ as a candidate solution to $(P)$ and of $x^2$ as a candidate solution to $(D)$.

The role of accuracy certificates in convex-concave saddle point problems stems from the following observation.

**Proposition A.2.** *Let $X_1, X_2$ be nonempty closed convex sets, $\Phi : X := X_1 \times X_2 \to \mathbf{R}$ be a locally Lipschitz continuous convex-concave function, and $F$ be the associated monotone vector field on $X$.*

*Let $\mathcal{I}_t = \{x_\tau = [x^1_\tau; x^2_\tau] \in X, F(x_\tau)\}_{\tau=1}^t$ be a t-step execution protocol associated with $(X, F)$ and $\lambda^t = \{\lambda^t_\tau\}_{\tau=1}^t$ be an associated accuracy certificate. Then $x^t := x^t(\mathcal{I}_t, \lambda^t) = [x^{1,t}; x^{2,t}] \in X$.*

*Assume, further, that $X_1' \subset X_1$ and $X_2' \subset X_2$ are closed convex sets such that*

$$x^t \in X' := X_1' \times X_2'. \tag{23}$$

*Then*

$$\epsilon_{\text{Sad}}(x^t | X_1', X_2', \Phi) = \sup_{x^2 \in X_2'} \Phi(x^{1,t}, x^2) - \inf_{x^1 \in X_1'} \Phi(x^1, x^{2,t}) \leq \text{Res}(X' | \mathcal{I}_t, \lambda^t). \tag{24}$$

*In addition, setting $\widetilde{\Phi}(x^1) = \sup_{x^2 \in X_2'} \Phi(x^1, x^2)$, for every $\bar{x}^1 \in X_1'$ we have*

$$\widetilde{\Phi}(x^{1,t}) - \widetilde{\Phi}(\bar{x}^1) \leq \widetilde{\Phi}(x^{1,t}) - \Phi(\bar{x}^1, x^{2,t}) \leq \text{Res}(\{\bar{x}^1\} \times X_2' | \mathcal{I}_t, \lambda^t). \tag{25}$$

*In particular, when the problem $\text{Opt} = \min_{x^1 \in X_1'} \widetilde{\Phi}(x^1)$ is solvable with an optimal solution $x^1_*$, we have*

$$\widetilde{\Phi}(x^{1,t}) - \text{Opt} \leq \text{Res}(\{x^1_*\} \times X_2' | \mathcal{I}_t, \lambda^t). \tag{26}$$

*Proof.* The inclusion $x^t \in X$ is clear. For every set $Y \subset X$ we have

$$
\begin{aligned}
&\forall [p;q] \in Y: \\
&\mathrm{Res}(Y|\mathcal{I}_t, \lambda^t) \geq \sum_{\tau=1}^{t} \lambda_\tau^t \left[ \langle F_1(x_\tau^1), x_\tau^1 - p \rangle + \langle F_2(x_\tau^2), x_\tau^2 - q \rangle \right] \\
&\geq \sum_{\tau=1}^{t} \lambda_\tau^t \left[ [\Phi(x_\tau^1, x_\tau^2) - \Phi(p, x_\tau^2)] + [\Phi(x_\tau^1, q) - \Phi(x_\tau^1, x_\tau^2)] \right] \\
&\text{[by the origin of } F \text{ and since } \Phi \text{ is convex-concave]} \\
&= \sum_{\tau=1}^{t} \lambda_\tau^t \left[ \Phi(x_\tau^1, q) - \Phi(p, x_\tau^2) \right] \geq \Phi(x^{1,t}, q) - \Phi(p, x^{2,t}) \\
&\qquad \text{[by origin of } x^t \text{ and since } \Phi \text{ is convex-concave]}
\end{aligned}
$$

Thus, for every $Y \subset X$ we have

$$
\sup_{[p;q] \in Y} \left[ \Phi(x^{1,t}, q) - \Phi(p, x^{2,t}) \right] \leq \mathrm{Res}(Y|\mathcal{I}_t, \lambda^t). \tag{27}
$$

Now assume that Condition (23) is satisfied. Setting $Y = X' := X_1' \times X_2'$, and recalling what $\epsilon_{\mathrm{Sad}}$ is, (27) yields (24). With $Y = \{\bar{x}^1\} \times X_2'$ (27) yields the second inequality in (25); the first inequality in (25) is clear since $x^{2,t} \in X_2'$. $\qquad\square$

## B Theoretical analysis of composite Mirror Prox with inexact proximal mappings

We restate the Theorem 3.1 and state Corollary B.1, and prove both results below. The theoretical convergence rate established in Theorem 3.1 and Corollary B.1 extends the previous result established in Corollary 3.1 in [11] for CMP with exact prox-mappings. Indeed, when exact prox-mappings are used, we recover the result of [11]. When inexact prox-mappings are used, the errors due to the inexactness of the prox-mappings accumulates and is reflected in the bound (29) and (37).

**Theorem 3.1.** *Assume that the sequence of step-sizes* $(\gamma_t)$ *in the CMP algorithm satisfy*

$$
\sigma_t := \gamma_t \langle F_u(\hat{u}^t) - F_u(u^t), \hat{u}^t - u^{t+1} \rangle - V_{\hat{u}^t}(u^{t+1}) - V_{u^t}(\hat{u}^t) \leq \gamma_t^2 M^2, \quad t = 1, 2, \ldots, T. \tag{28}
$$

*Then, denoting* $\Theta[X] = \sup_{[u;v] \in X} V_{u^1}(u)$, *for a sequence of inexact prox-mappings with inexactness* $\epsilon_t \geq 0$, *we have*

$$
\epsilon_{\mathrm{VI}}(\bar{x}_T|X, F) := \sup_{x \in X} \langle F(x), \bar{x}_T - x \rangle \leq \frac{\Theta[X] + M^2 \sum_{t=1}^{T} \gamma_t^2 + 2 \sum_{t=1}^{T} \epsilon_t}{\sum_{t=1}^{T} \gamma_t}. \tag{29}
$$

**Remarks** Note that the assumption on the sequence of step-sizes $(\gamma_t)$ is clearly satisfied when $\gamma_t \leq (\sqrt{2}L)^{-1}$. When $M = 0$, it is satisfied as long as $\gamma_t \leq L^{-1}$.

*Proof.* The proof builds upon and extends the proof in [11]. For all $u, u', w \in U$, we have the so-called three-point identity

$$
\langle V_u'(u'), w - u' \rangle = V_u(w) - V_{u'}(w) - V_u(u'). \tag{30}
$$

For $x = [u;v] \in X$, $\xi = [\eta;\zeta]$, $\epsilon \geq 0$, let $[u';v'] \in P_x^\epsilon(\xi)$. By definition, for all $[s;w] \in X$, the inequality holds

$$
\langle \eta + V_u'(u'), u' - s \rangle + \langle \zeta, v' - w \rangle \leq \epsilon,
$$

which by (30) implies that

$$
\langle \eta, u' - s \rangle + \langle \zeta, v' - w \rangle \leq \langle V_u'(u'), s - u' \rangle + \epsilon = V_u(s) - V_{u'}(s) - V_u(u') + \epsilon. \tag{31}
$$

When applying (31) with $\epsilon = \epsilon_t$, $[u;v] = [u^t;v^t] = x^t$, $\xi = \gamma_t F(x^t) = [\gamma_t F_u(u^t); \gamma_t F_v]$, $[u';v'] = [\hat{u}^t; \hat{v}^t] = y^t$, and $[s;w] = [u^{t+1}; v^{t+1}] = x^{t+1}$, we obtain

$$
\gamma_t [\langle F_u(u^t), \hat{u}^t - u^{t+1} \rangle + \langle F_v, \hat{v}^t - v^{t+1} \rangle] \leq V_{u^t}(u^{t+1}) - V_{\hat{u}^t}(u^{t+1}) - V_{u^t}(\hat{u}^t) + \epsilon_t; \tag{32}
$$

and applying (31) with $\epsilon = \epsilon_t$, $[u;v] = x^t$, $\xi = \gamma_t F(y^t)$, $[u';v'] = x^{t+1}$, and $[s;w] = z \in X$ we get

$$
\gamma_t [\langle F_u(\hat{u}^t), u^{t+1} - s \rangle + \langle F_v, v^{t+1} - w \rangle] \leq V_{u^t}(s) - V_{u^{t+1}}(s) - V_{u^t}(u^{t+1}) + \epsilon_t. \tag{33}
$$

Adding (33) to (32), we obtain for every $z = [s; w] \in X$

$$\gamma_t \langle F(y^t), y^t - z \rangle = \gamma_t[\langle F_u(\widehat{u}^t), \widehat{u}^t - s \rangle + \langle F_v, \widehat{v}^t - w \rangle]$$
$$\leq V_{u^t}(s) - V_{u^{t+1}}(s) + \sigma_t + 2\epsilon_t \,, \tag{34}$$

with

$$\sigma_t := \gamma_t \langle F_u(\widehat{u}^t) - F_u(u^t), \widehat{u}^t - u^{t+1} \rangle - V_{\widehat{u}^t}(u^{t+1}) - V_{u^t}(\widehat{u}_t) \,.$$

Due to the strong convexity, with modulus 1, of $V_u(\cdot)$ w.r.t. $\|\cdot\|$, we have for all $u, \widehat{u}$

$$V_u(\widehat{u}) \geq \frac{1}{2}\|u - \widehat{u}\|^2 \,.$$

Therefore,

$$\begin{aligned}
\sigma_t &\leq \gamma_t\|F_u(\widehat{u}^t) - F_u(u^t)\|_*\|\widehat{u}^t - u^{t+1}\| - \tfrac{1}{2}\|\widehat{u}^t - u^{t+1}\|^2 - \tfrac{1}{2}\|u^t - \widehat{u}^t\|^2 \\
&\leq \tfrac{1}{2}\left[\gamma_t^2\|F_u(\widehat{u}^t) - F_u(u^t)\|_*^2 - \|u^t - \widehat{u}^t\|^2\right] \\
&\leq \tfrac{1}{2}\left[\gamma_t^2[M + L\|\widehat{u}^t - u^t\|]^2 - \|u^t - \widehat{u}^t\|^2\right] \,,
\end{aligned}$$

where the last inequality follows from Assumption **A**.3. Note that $\gamma_t L < 1$ implies that

$$\gamma_t^2[M + L\|\widehat{u}^t - u^t\|]^2 - \|\widehat{u}^t - u^t\|^2 \leq \max_r \left[\gamma_t^2[M + Lr]^2 - r^2\right] = \frac{\gamma_t^2 M^2}{1 - \gamma_t^2 L^2}.$$

Let us assume that the (nonnegative) step-sizes $(\gamma_t)$ are chosen so that (28) holds, that is $\sigma_t \leq \gamma_t^2 M^2$. It is indeed the case when $0 < \gamma_t \leq 1/\sqrt{2}L$; when $M = 0$, we can take also $\gamma_t \leq 1/L$. Summing up inequalities (34) over $t = 1, 2, ..., t$, and taking into account that $V_{u^{t+1}}(s) \geq 0$, we finally conclude that for all $z = [s; w] \in X$,

$$\sum_{t=1}^{T} \lambda_T^t \langle F(y^t), y^t - z \rangle \leq \frac{V_{u^1}(s) + M^2 \sum_{t=1}^{T} \gamma_t^2 + 2\sum_{t=1}^{T} \epsilon_t}{\sum_{t=1}^{T} \gamma_t}, \text{ where } \lambda_T^t = \left(\sum_{i=1}^{T} \gamma_i\right)^{-1}\gamma_t \,.$$

$\square$

**Corollary B.1.** *Assume further that $X = X_1 \times X_2$, and let $F$ be the monotone vector field associated with the saddle point problem*

$$\text{SadVal} = \min_{x^1 \in X_1} \max_{x^2 \in X_2} \Phi(x^1, x^2), \tag{35}$$

*two induced convex optimization problems*

$$\begin{array}{lll}
\text{Opt}(P) &= \min_{x^1 \in X_1} \left[\overline{\Phi}(x^1) = \sup_{x^2 \in X_2} \Phi(x^1, x^2)\right] & (P) \\
\text{Opt}(D) &= \max_{x^2 \in X_2} \left[\underline{\Phi}(x^2) = \inf_{x^1 \in X_1} \Phi(x^1, x^2)\right] & (D)
\end{array} \tag{36}$$

*with convex-concave locally Lipschitz continuous cost function $\Phi$. In addition, assuming that problem $(P)$ in (36) is solvable with optimal solution $x_*^1$ and denoting by $\bar{x}_T^1$ the projection of $\bar{x}_T \in X = X_1 \times X_2$ onto $X_1$, we have*

$$\overline{\Phi}(\bar{x}_T^1) - \text{Opt}(P) \leq \left[\sum_{t=1}^{T} \gamma_t\right]^{-1} \left[\Theta[\{x_*^1\} \times X_2] + M^2 \sum_{t=1}^{T} \gamma_t^2 + 2\sum_{t=1}^{T} \epsilon_t\right]. \tag{37}$$

## C Theoretical analysis of composite conditional gradient

### C.1 Convergence rate

The CCG algorithm enjoys a convergence rate in $O(t^{-(\kappa-1)})$ in the evaluations of the function $\phi^+$, and the accuracy certificates $(\delta_t)$ enjoy the same rate $O(t^{-(\kappa-1)})$ as well, for solving problems of type (10).

**Proposition 3.1.** *Denote $D$ the $\|\cdot\|$-diameter of $U$. When solving problems of type (10), the sequence of iterates $(x^t)$ of CCG satisfies*

$$\phi^+(x^t) - \min_{x \in X} \phi^+(x) \leq \frac{2L_0 D^\kappa}{\kappa(3 - \kappa)} \left(\frac{2}{t + 1}\right)^{\kappa - 1}, \, t \geq 2 \tag{38}$$

*In addition, the accuracy certificates $(\delta_t)$ satisfy*

$$\min_{1 \leq s \leq t} \delta_s \leq O(1) L_0 D^\kappa \left(\frac{2}{t + 1}\right)^{\kappa - 1}, \, t \geq 2 \tag{39}$$

## C.2 Proof of Proposition 3.1

Let us define $\epsilon_t = \phi^+(x^t) - \min_{x \in X} \phi^+(x)$.

$\mathbf{1^0.}$ The projection of $X$ onto $E_u$ is contained in $U$, whence

$$\|u[\nabla\phi(u^t)] - u^t\| \le D, \forall t = 1, 2, \dots.$$

This observation, due to the structure of $\phi^+$, implies that whenever $x, x' \in X$ and $\gamma \in [0, 1]$, we have

$$\phi^+(x + \gamma(x^+ - x)) \le \phi^+(x) + \gamma\langle\nabla\phi^+(x), x' - x\rangle + \frac{L_0 D^\kappa}{\kappa}\gamma^\kappa. \tag{40}$$

Setting $x_+^t = x^t + \gamma_t(x[\nabla\phi(u^t)] - x^t)$ and $\gamma_t = 2/(t+1)$, we have

$$\epsilon_{t+1} \le \phi^+(x_+^t) - \min_{x \in X} \phi^+(x) \tag{41}$$

$$\le \epsilon_t + \gamma_s\langle\nabla\phi(x^t), x[\nabla\phi^+(x^t)] - x\rangle + \frac{L_0 D^\kappa}{\kappa}\gamma_t^\kappa \tag{42}$$

$$= \epsilon_t - \gamma_t\delta_t + \frac{L_0 D^\kappa}{\kappa}\gamma_t^\kappa, \tag{43}$$

whence, due to $\delta_t \ge \epsilon_t \ge 0$,

$$(i) \quad \epsilon_{t+1} \le (1 - \gamma_t)\epsilon_t + \frac{L_0 D^\kappa}{\kappa}\gamma_t^\kappa, \ t = 1, 2, \dots,$$

$$(ii) \quad \gamma_s\delta_s \le \epsilon_s - \epsilon_{s+1} + \frac{L_0 D^\kappa}{\kappa}\gamma_s^\kappa, \ s = 1, 2, \dots \tag{44}$$

$\mathbf{2^0.}$ Let us prove (38) by induction on $s \ge 2$. By (44.i) and due to $\gamma_1 = 1$ we have

$$\epsilon_2 \le \frac{L_0 D^\kappa}{\kappa}. \tag{45}$$

Whence, due to $\gamma_2 = 2/3$ and $1 < \kappa \le 2$, we get

$$\epsilon_2 \le \frac{2L_0 D^\kappa}{\kappa(3 - \kappa)}\gamma_2^{\kappa-1}. \tag{46}$$

Now, assume that, for some $t \ge 2$

$$\epsilon_t \le \frac{2L_0 D^\kappa}{\kappa(3 - \kappa)}\gamma_t^{\kappa-1}. \tag{47}$$

Then, invoking (44.i),

$$\epsilon_{t+1} \le \frac{2L_0 D^\kappa}{\kappa(3 - \kappa)}\gamma_t^{\kappa-1}(1 - \gamma_t) + \frac{L_0 D^\kappa}{\kappa}\gamma_t^\kappa$$

$$\le \frac{2L_0 D^\kappa}{\kappa(3 - \kappa)}\left[\gamma_t^{\kappa-1} - \frac{\kappa - 1}{2}\gamma_t^\kappa\right]$$

$$\le \frac{2L_0 D^\kappa}{\kappa(3 - \kappa)}2^{\kappa-1}\left[(t + 1)^{1-\kappa} + (1 - \kappa)(t + 1)^{-\kappa}\right]$$

Therefore, by convexity of $(t + 1)^{1-\kappa}$ in $t$

$$\epsilon_{t+1} \le \frac{2L_0 D^\kappa}{\kappa(3 - \kappa)}2^{\kappa-1}(t + 2)^{1-\kappa} = \frac{2L_0 D^\kappa}{\kappa(3 - \kappa)}\gamma_{t+1}^{\kappa-1}$$

The induction is completed.

$\mathbf{3^0.}$ To prove (39), given $s \ge 2$, let $t_- = \text{Ceil}(\max[2, t/2])$. Summing up inequalities (44.ii) over $t_- \le s \le t$, we get

$$\left(\min_{1 \le s \le t} \delta_s\right)\sum_{s=t_-}^t \gamma_s \le \sum_{s=t_-}^t \gamma_s\delta_s \le \epsilon_{t_-} - \epsilon_{t+1} + \frac{L_0 D^\kappa}{2}\sum_{s=t_-}^t \gamma_s^\kappa \le O(1)L_0 D^\kappa\gamma_t^{\kappa-1}$$

and $\sum_{s=t_-}^t \gamma_s \ge O(1)$, and (39) follows.

# D Semi-Proximal Mirror-Prox

## D.1 Theoretical analysis for Semi-Proximal Mirror-Prox

We first restate Proposition 3.2 and provide the proof below.

**Proposition 3.2.** *Under the assumption* $(A.1) - (A.4)$ *and* $(S.1) - (S.3)$ *with* $M = 0$, *for the outlined algorithm to return an* $\epsilon$-*solution to the variational inequality* $VI(X, F)$, *the total number of Mirror Prox steps required does not exceed* $O\left(\frac{L\Theta[X]}{\epsilon}\right)$, *and the total number of calls to the Linear Minimization Oracle does not exceed*

$$\mathcal{N} = O(1) \left(\frac{L_0 L^\kappa D^\kappa}{\epsilon^\kappa}\right)^{\frac{1}{\kappa-1}} \Theta[X].$$

*In particular, if we use Euclidean proximal setup on* $U_2$ *with* $\omega_2(\cdot) = \frac{1}{2}\|x_2\|^2$, *which leads to* $\kappa = 2$ *and* $L_0 = 1$, *then the number of LMO calls does not exceed* $\mathcal{N} = O(1) \left(L^2 D^2(\Theta[X_1] + D^2)\right)/\epsilon^2$.

*Proof.* Let us fix $N$ as the number of Mirror prox steps, and since $M = 0$, from Theorem 3.1, the efficiency estimate of the variational inequality implies that

$$\epsilon_{\mathrm{VI}}(\bar{x}^N | X, F) \leq \frac{L(\Theta[X] + 2\sum_{t=1}^N \epsilon_t)}{N}.$$

Let us fix $\epsilon_t = \frac{\Theta[X]}{2N}$ for each $t = 1, \ldots, N$, then from Proposition 3.1, it takes at most $s = O(1)(\frac{L_0 D^\kappa N}{\Theta[X]})^{1/(\kappa-1)}$ LMO oracles to generate a point such that $\Delta_s \leq \epsilon_t$. Moreover, we have

$$\epsilon_{\mathrm{VI}}(\bar{x}^N | X, F) \leq 2\frac{L\Theta[X]}{N}.$$

Therefore, to ensure $\epsilon_{\mathrm{VI}}(\bar{x}^N | X, F) \leq \epsilon$ for a given accuracy $\epsilon > 0$, the number of Mirror Prox steps $N$ is at most $O(\frac{L\Theta[X]}{\epsilon})$ and the number of LMO calls on $X_2$ needed is at most

$$\mathcal{N} = O(1)\left(\frac{L_0 L^\kappa D^\kappa}{\epsilon^\kappa}\right)^{1/(\kappa-1)} \Theta[X].$$

In particular, if $\kappa = 2$ and $L_0 = 1$, this quantity can be reduced to $\mathcal{N} = O(1)\frac{L^2 D^2 \Theta[X]}{\epsilon^2}$. □

## D.2 Discussion of Semi-Proximal Mirror-Prox

The proposed Semi-Proximal Mirror-Prox algorithm enjoys the *optimal complexity bounds*, i.e. $O(1/\epsilon^2)$, in the number of calls to linear minimization oracle. Furthermore, Semi-Proximal Mirror-Prox generalizes previously proposed approaches and improves upon them in special cases of problem (3).

When there is no regularization penalty, Semi-Proximal Mirror-Prox is more general than previous algorithms for solving the corresponding constrained non-smooth optimization problem. Semi-Proximal Mirror-Prox does not require assumptions on favorable geometry of dual domains $Z$ or simplicity of $\psi(\cdot)$ in (2). When the regularization is simply a norm (with no operator in front of the argument), Semi-Proximal Mirror-Prox is competitive with previously proposed approaches [15, 24] based on smoothing techniques.

When the regularization penalty is non-trivial, Semi-Proximal Mirror-Prox is the first proximal-free or conditional-gradient-type optimization algorithm, up to our knowledge.

# E Numerical experiments and implementation details

## E.1 Robust collaborative filtering: $\ell_1$-empirical risk +nuclear norm

We consider the collaborative filtering problem, with a nuclear-norm regularisation penalty and an $\ell_1$-empirical risk function:

$$\min_x \frac{1}{|E|} \sum_{(i,j)\in E} |x_{ij} - b_{ij}| + \lambda\|x\|_{\mathrm{nuc}}. \tag{48}$$

**Competing algorithms.** We compare the above three candidate algorithm. The smoothed problem for Semi-SPG and Smooth-CG in this case becomes

$$\min_{x,v:\|x\|_{\mathrm{nuc}}\leq v} f^{\gamma}(x) + \lambda v, \text{ where } f^{\gamma}(x) = \max_{\|y\|_{\infty}\leq 1}\left\{\frac{1}{|E|}\sum_{(i,j)\in E}(x_{ij}-b_{ij})y_{ij} - \frac{\gamma}{2}\|y\|_2^2\right\}. \quad (49)$$

Note that in this case, for Smooth-CG, the update of $x$ at step $t$ requires solving reoptimization problem

$$\min_{\theta_1,\dots,\theta_t} f^{\gamma}(\sum_{i=1}^t \theta_i u_i v_i^T) + \lambda \sum_{i=1}^t \theta_i \quad (50)$$

which requires computing the full matrix representation for the gradient. For large $t$ and large-scale problems, the computation cost for re-optimization is no longer negligible. However, the Semi-MP and Semi-SPG do not suffer from this limitation since the conditional gradient routines are called for simple quadratic subproblems. For this particular example, we implement the Semi-MP slightly different from the above scheme. We solve the following saddle point reformulation with properly selected $\rho$,

$$\min_{\substack{x,y,v_1,v_2: \\ v_1 \geq \|x\|_{\mathrm{nuc}}, v_2 \geq \|y\|_1}} \max_{\|w\|_2 \leq 1} v_2 + \lambda v_1 + \rho\langle \mathcal{A}x - b - y, w\rangle \quad (51)$$

where we use $\mathcal{A}$ to denote the operator $\frac{1}{|E|}P_E$. The semi-structured variational inequality Semi-VI $(X, F)$ associated with the above saddle point problem is given by $X = \{[u = (x, y, w); v = (v_1.v_2)] : \|x\|_{\mathrm{nuc}} \leq v_1, \|y\|_1 \leq v_2, \|w\|_2 \leq 1\}$ and $F = [F_u(u); F_v] = [\rho\mathcal{A}w; -\rho w; \rho(y - \mathcal{A}x + b); \lambda; 1]$. The subdomain $X_1 = \{(y, w, v_2) : \|y\|_1 \leq v_2, \|w\|_2 \leq 1\}$ is given by full-prox setup and the subdomain $X_2 = \{(x; v_1) : \|x\|_{\mathrm{nuc}} \leq v_1\}$ is given by LMO. By setting both the distance generating functions as the Euclidean distance, the update of $w$ reduces to the gradient step, the update of $y$ reduces to the soft-thresholding operator, and the update of $x$ is given by the composite conditonal gradient routine. In our experiment, the factor $\rho$ is updated adaptively as follows. We start with a small $\rho = 1e-3$ and increase it with a factor of 3 until when enforcing $y = x$ does not increase the objective function value. We set the stepsizes $\gamma_t$ along the iterations using line-search. All in all, the Semi-Proximal Mirror-Prox algorithm (Semi-MP) is fully automatic, and does not require tuning of any parameter.

We run the above three algorithms on the the small and medium MovieLens datasets. The small-size dataset consists of 943 users and 1682 movies with about 100K ratings,while the medium-size dataset consists of 3952 users and 6040 movies with about 1M ratings. We follow [24] to set the regularization parameters. We randomly pick 80% of the entries to build the training dataset, and compute the normalized mean absolute error (NMAE) on the remaining test dataset. For Smooth-CG, we carry out the algorithm with different smoothing parameters, ranging from $\{1e-3, 1e-2, 1e-1, 1e0\}$ and select the one with the best performance. For the Semi-SPG algorithm, we adopt the best smoothing parameter found in Smooth-CG. We use two different strategies to control the number of LMO calls at each iteration, i.e. the accuracy of the proximal mapping for both Semi-SPG and Semi-MP, which are a) fixed inner CG steps and b) decaying $\epsilon_t = c/t$ as the theory suggested. We report in Fig. 2 and Fig. 3 the performance of each algorithm under different choice of parameters and the overall comparison of objective value and NMAE on test data in Fig. 4.

Figure 2: Robust collaborative filtering on MovieLens 100K: objective function vs elapsed time. From left to right: (a) Semi-MP; (b) Semi-SPG ; (c) Smooth-CG; (d) best of three algorithms.

Figure 3: Robust collaborative filtering on MovieLens 1M: objective function vs elasped time. From left to right: (a) Semi-MP; (b) Semi-SPG ; (c) Smooth-CG; (d) best of three algorithms.

Figure 4: Robust collaborative filtering on Movie Lens: objective function and test NMAE against elapsed time. From left to right: (a) MovieLens 100K objective; (b) MovieLens 100K test NMAE; (c) MovieLens 1M objective; (d) MovieLens 1M test NMAE.

In Fig. 2 and Fig. 3, we can see that using fixed inner CG steps sometimes achieve comparable performance as using the decaying epsilon $\epsilon_t$. In Fig. 4, we can see that Semi-MP clearly outperforms Smooth-CG, while it is competitive with Semi-SPG. In the large-scale setting, Semi-MP achieves better objective as well as test NMAE compared to Smooth-CG.

### E.2  Link prediction: hinge loss + $\ell_1$-norm + nuclear norm

We consider the following model for the link prediction problem,

$$\min_{x \in \mathbf{R}^{m \times n}} \frac{1}{|E|} \sum_{(i,j) \in E} \max\left(1 - (b_{ij} - 0.5)x_{ij}, 0\right) + \lambda_1 \|x\|_1 + \lambda_2 \|x\|_{\text{nuc}} \tag{52}$$

This example is more complicated than the previous two examples since it has not only one nonsmooth loss function but also two regularization terms. Applying the smoothing-CG or Semi-SPG would require to build two smooth approximations, one for hinge loss term and one for $\ell_1$ norm term. Therefore, we consider another alternative approach, Semi-LPADMM, where we apply the linearized preconditioned ADMM algorithm by solving proximal mapping through conditional gradient routines. Up to our knowledge, ADMM with early stopping is not well-analyzed in literature, but intuitively as long as the accumulated error is controlled sufficiently, the variant will converge.

Figure 5: Link prediction on Wikivote: objective function value against the LMO calls. From left to right: (a)Wikivote(1024) with fixed inner steps; (b) Wikivote(1024) with $\epsilon_t = c/t$; (c) Wikivote(full)

We conduct experiments on a binary social graph data set called Wikivote, which consists of 7118 nodes and 103,747 edges. Since the computation cost of these two algorithms mainly come from the LMO calls, we present in below the performance in terms of number of LMO calls. For the first set of experiments, we select top 1024 highest degree users from Wikivote and run the two algorithms on this small dataset with different strategies for the inner LMO calls.

In Fig. 5, we observe that the Semi-MP is less sensitive to the inner accuracies of prox-mappings compared to the ADMM variant, which sometimes stop progressing if the prox mapping of early iterations are not solved with sufficient accuracy. Another observation is that in this example, the second strategy, which essentially saves the use of LMOs, works better in the long run than using fixed number of LMOs. The results indicate again on the full dataset again indicates that our algorithm performs better than the semi-proximal variant of ADMM algorithm.