[Reviews · NeurIPS 2015]

Submitted by Assigned_Reviewer_1

A 'prox-free' conditional gradient algorithm is proposed for a general class of problems, exploiting conjugate representation. I enjoyed reading the presentation and development - the exposition of the saddle point reformulation, and the explicit variational inequality for the problem class, makes it especially clear what the main idea is, and how it fits into the context of existing methods.

One thing I hope the authors will clarify in the final version is as follows. They replace y = Bx with an added penalty term \rho ||y - Bx||, and the variational inequality proceeds from this development. Do they guarantee y = Bx holds at the solution, by way of exact penalization? There is a brief note in the appendix on how rho is adjusted - it seems rho is adjusted on the fly, but exactly how it is done, and whether you guarantee y = Bx when you're done was not clear.

******

Thanks for clarifying in the rebuttal.
Summary: Authors present a prox-free algorithm for a general class of problems. The development is well motivated and clear, contributions clearly stated, and numerical results are interesting. I think this paper will be of interest to the community.

Submitted by Assigned_Reviewer_2

After the rebuttal:

The theory developed for inexact composite mirror prox is a trivial modification of the proof of Theorem 3.1 in [10]. Authors do not hide this fact. In the end, idea-wise this paper is close to [14] and technically it does not appear to be too much involved.

To me, the most interesting point of this work is that the objective function can be doubly non-smooth, in the sense f can also be non-smooth. On that front, how does this work relate to

http://arxiv.org/abs/1502.03123

A. Yurtsever, Q. Tran-Dinh, and V. Cevher "Universal Primal-Dual Proximal-Gradient Methods"

which appears also to solve non-smooth constrained problems with Fenchel-type oracles?

============ This paper proposes a new first order optimization algorithm to solve the composite problem (1) under assumptions A1-A4,S1-S3 (see the paper), which can be considered as a variant of composite mirror prox method (CMP), where the prox-operator are solved inexactly using composite conditional gradient algorithm (CCG). This paper lacks the references for some related recent works (e.g. [1] and [2]).

Technically, the authors show how the inexactness in the CMP iterates accumulate in the dual gap function, they explain how we should choose the accuracy input for CCG subroutines and provide the corresponding worst case complexity analysis, which can be considered as the novel part of the work.

However, from an intellectual point of view, the idea of solving prox-operator approximately reminds us [2] and it is not clear how CCG subroutine that is presented in the paper is any different from the algorithms presented in [1]. (Note that the assumption (13) is a consequence of the Holder continuity assumption in [1]).

Considering the presence of [1] and [2], the significance of this work becomes questionable.

Finally, we provide below a list of minor issues and typos for the authors: 1. US English and UK English are used inconsistently. In some parts of the text 'minimization' and 'optimization' are used whereas in some other parts these are written as 'minimisation' and 'optimisation'. 2. line 91 - point is missing at line after reference [13]. 3. line 141 - 'be' missing: can 'be' much cheaper. 4. line 214 - 'of' missing: in terms 'of' this dual gap function ... 5. line 217 - typo : fro - for 6. line 247 - there is an extra parenthesis ')' at the end of line. 7. line 256 - typo: proc-mapping - prox-mapping 8. line 271 - L < inf should be L_0 < inf / or L_0 in eq (13) should be L 9. line 278 - typo: given 'on' input - given 'an' input 10. line 324 - repetition: we assume that 'we we' have 11. line 365 - Assumptions (B.1)-(B.3) should be replaced by Assumptions (S.1)-(S.3) 12. line 776 - We shall compare and 'compare' - We shall compare and 'contrast'

[1] Y. Nesterov, Complexity Bounds for Primal-dual Methods Minimizing the Model of Objective Function, CORE Discussion Paper, 2015. [2] G. Lan, Gradient Sliding for Composite Optimization, arxiv:1406.0919v2.
Summary: This paper proposes a new first order optimization algorithm which is in fact a variant of composite mirror prox method (CMP), where the prox-operator are solved inexactly using composite conditional gradient algorithm (CCG). The idea of approximating the prox-operator reminds us [2] and it is not clear how CCG subroutine that is presented in the paper is different from the algorithms presented in [1]. In the presence of [1] and [2], the significance of this work is questionable.

(see Comments to authors part for references).

Submitted by Assigned_Reviewer_3

The paper is very difficult to follow, because there is a lot of notations. The work is very difficult, and use the variational inequality framework too derive the algorithm. Variational inequality is a very general framework, but I think difficult to understand by the non specialist. An exemplification of the algorithm on a simple example (as the one use for the experiments) would be very helpful.
Summary: An optimal algorithm for non smooth convexe optimization, with "analysis" prior. Very difficult too read for the non specialists.

Submitted by Assigned_Reviewer_4

The regularized risk minimization problem, where both the regularizer as well as the empirical risk terms both could potentially be non-smooth, is considered. The paper provides a novel Conditional gradient method for this setting. The key idea is to write down the original problem in a saddle point form using Fenchel-dual representation of the empirical term and devising composite versions of proximal and conditional gradient methods. The convergence rate is shown, which in general, cannot be improved.

The algorithm as well as its analysis are interesting. The proposed algorithm not only generalizes existing methods to (1), but also seems to improve in the special cases.

However, currently it seems that the method is at best comparable to smoothing+acceleratedgradient. Simulations showing otherwise will be more convincing.
Summary: An overall interesting and well-written paper that presents the first conditional gradient based algorithm for doubly non-smooth problems of the form (1).

Author Feedback
Author rebuttal: We appreciate all reviewers' careful reading and insightful comments. We are very grateful to the overall positive assessment of the novelty and significance of our work. In what follows, we address the reviewers' concerns and questions.

To Reviewer 1
1. Do they guarantee y=Bx holds at the solution, by way of exact penalization?

The short answer is yes. The saddle point reformulation does not require y=Bx, so the algorithm does not necessarily produce a solution (x,y) that satisfies the equation. Yet, one can show that when the penalty coefficient is sufficiently large (decided by some Lipschitz constant), the "corrected" solution (x,Bx) (simply setting y=Bx) is at least as good as the pair (x,y). We will add the details to clarify this point in the final version.

2. it seems rho is adjusted on the fly, but exactly how it is done, and whether you guarantee y=Bx when you're done was not clear.

The answer is closed related the the above one. In principle, if the corresponding Lipschitz constant is known, we can simply set that as the value of rho and the above 'correction' step will enforce y=Bx without contaminating the solution. Here is how rho can be adjusted in practice. We start at the beginning with a small rho. At each iteration, we do the correction step and pass the solution (x^t,y^t) to (x^t,Bx^t) and increase the current rho in a fixed ratio whenever the corrected solution does not reduce a 'significant' amount of the objective value. We will add the details about this in the final version.

To Reviewer 3
1. However, currently it seems that the method is at best comparable to smoothing+accelerated gradient. Simulations showing otherwise will be more convincing.

In principle, our method achieves the same computation complexity as the semi-proximal variant of accelerated gradient after smoothing (denoted as Semi-SPG) if it is applicable. So, we are not expecting our algorithm to beat Semi-SPG in those situations. In fact, in the matrix completion simulation, we actually observe that our method performs better than Semi-SPG.
The main strength of our algorithm is that it works for a broader class of problems than Semi-SPG, e.g. the link prediction with both sparse and low-rank penalty terms, also discussed in the experiments.

To Reviewer 8
1. This paper lacks the references for some related recent works (e.g. [1, Nesterov 2015] and [2, Lan 2014]).

We have included Lan's conditional gradient sliding paper in the reference [14], which we believe is more relevant than [2].
We will include [1] in the final version.

2. The idea of solving prox-operator approximately reminds us [2] and it is not clear how CCG subroutine that is presented in the paper is any different from the algorithms presented in [1].

Solving subproblems approximately via other optimization routines (e.g. using block coordinate descent [Schmidtz 2011], accelerated gradient descent [Lan 2014], conditional gradient [Zhou and Lan 2014], etc) has been explored in various settings.
But not all algorithms support inexact prox-operators. To our best knowledge, such an approach, along with theoretical analysis, for inexact mirror prox using conditional gradient routine is absolutely novel. The CCG routine, as we stated in the text, is especially tailored for the smooth semi-linear problems. Compared to [1]: the v-component does not necessarily have to be induced from a simple nonsmooth function, so it is more general.

3. Considering [1] and [2], the significance of this work becomes questionable.

Our contributions might have been overlooked here. We restate and underline them below:
- we have developed theories for inexact (composite) Mirror-Prox algorithm;
- we have proposed a new approach that solves a broad class of variational inequalities, which not only generalizes existing methods but also improves upon them in many cases;
- we provide theoretical analysis of the approach which exhibits optimal complexity bounds both in terms of first-order oracles and linear minimization oracles;
- we provide promising numerical results on real-world datasets that shows great interest of the approach in practice.

4. A list of minor typos.
Corrected.

To Reviewer 9
1. Variational inequality is a very general framework...an exemplification of the algorithm on a simple example would be very helpful.

We understand the variational inequality could be difficult to understand by non-specialist. It is indeed our intention to introduce it to the machine learning community because we believe it is extremely important to this community in lots of aspects. We have illustrated the algorithm on all three examples coming from the experiments by explicitly providing the monotone operators and proximal setups in the appendix. We will introduce these examples earlier in the text to exemplify and illustrate the approach.